# ON INFORMATION DROPPING AND OVERSMOOTHING IN GRAPH NEURAL NETWORKS

## ABSTRACT

Graph Neural Networks (GNNs) are widespread in graph representation learning. *Random dropping* approaches, notably DropEdge and DropMessage, claim to alleviate the key issues of overfitting and oversmoothing by randomly removing elements of the graph representation. However, their effectiveness is largely unverified. In this work, we show empirically that they have a limited effect in reducing oversmoothing at test time due to their training time exclusive nature. We show that DropEdge in particular can be seen as a form of training data augmentation, and its benefits to model generalization are not strictly related to oversmoothing, suggesting that in practice, the precise link between oversmoothing and test time performance is more nuanced. We additionally address the limitations of current dropping methods by *learning* to drop, and propose a new information-theoretic approach, which performs dropping during message passing by optimizing an information bottleneck.

## 1 INTRODUCTION

Graphs are pervasive in the real world, effectively representing complex relationships among various entities across a multitude of domains such as social media (Fan et al., 2019), finance (Bi et al., 2022), and biology Jumper et al. (2021). Graph neural networks (GNNs), as state-of-the-art tools for graph representation learning, have garnered significant interest in recent years (Kipf & Welling, 2017; Hamilton et al., 2017; Veličković et al., 2018). At the core of GNNs lies a message-passing schema, which allows each node to aggregate information from its neighboring nodes.

Despite rapid advances in GNNs, they still face critical challenges. In particular, *oversmoothing* occurs when representations of different nodes in a GNN become indistinguishable, as they aggregate information from neighbors recursively (Oono & Suzuki, 2020). This phenomenon hinders GNNs from effectively modeling higher-order dependencies from multihop neighbors and makes them more vulnerable to adversarial attacks (Li et al., 2018; Chen et al., 2019). Common approaches for mitigating oversmoothing include adding regularization terms based on measures of oversmoothing (Chen et al., 2019), and restricting the pairwise distances between nodes (Zhao & Akoglu, 2020).

Another widely used approach is based on the *random dropping* of information from the graph or its representation. Prominent examples include DropEdge (Rong et al., 2020) and DropMessage (Fang et al., 2023), which operate on the edge and message levels respectively. Notably, DropMessage has been recently proposed as a generalization of DropEdge. However, the impact of these techniques on oversmoothing and the precise link between their oversmoothing reduction and the benefits to model performance have not been thoroughly investigated.

In this paper, we investigate the extent to which DropEdge and DropMessage are able to mitigate oversmoothing. We show that at test time, both methods actually have a limited effect in reducing oversmoothing according to metrics such as *Dirichlet energy* and *mean average distance*. We hypothesise that DropEdge has a similar effect to training with data augmentation and demonstrate that its beneficial effects on model performance are highly conditional on the randomness used in dropping. We also observe that enabling random dropping at test time will considerably reduce oversmoothing, but this *does not translate to improved performance*, suggesting that minimizing oversmoothing by itself is insufficient. This motivates *Learn2Drop*. In contrast to traditional dropping mechanisms, which apply a uniform approach to information pruning and reduce oversmoothing in

a deterministic manner, Learn2Drop learns a mask over the messages each node receives, enabling test-time message dropping to be performed dynamically.

The foundation of Learn2Drop is rooted upon the information bottleneck principle (Tishby et al., 2000). The bottleneck seeks a representation $Z$ that is minimally informative about the input $X$, whilst simultaneously being maximally informative about the target $Y$. By balancing $I(X, Z)$ and $I(Z, Y)$, it allows task-irrelevant information to be discarded while preserving useful information, allowing the GNN to focus on the most salient features of the data. In a sense, this potentially allows the GNN to *learn* to reduce oversmoothing in an optimal way.

## 2 RELATED WORK

### 2.1 DROPOUT IN NEURAL NETWORKS

Dropout, as introduced by (Srivastava et al., 2014), stems from the notion that the random deactivation of certain units during training equates to training an ensemble of networks. This process effectively counters overfitting in various models, GNNs included. DropEdge (Rong et al., 2020) adopts a different strategy by randomly omitting a subset of edges from the input graph prior to the standard message-passing procedure. This operation only occurs during training. Given an input graph $G = (E, V)$, they remove $p|V|$ edges randomly, where $p \in (0, 1)$ is a user-defined parameter. The authors argue that this approach simultaneously addresses both overfitting and oversmoothing. DropNode (Feng et al., 2020) is a similar approach, in which nodes are randomly removed, although it does not specifically aim to address oversmoothing. DropMessage (Fang et al., 2023) is another dropping approach in which elements of the *message matrix* are randomly dropped during training.

### 2.2 CURRENT UNDERSTANDING OF OVERSMOOTHING

Many earlier works on oversmoothing have proposed practical techniques to alleviate it (Chen et al., 2019; Zhao & Akoglu, 2020; Rong et al., 2020; Chen et al., 2020). Recently there has been a greater focus investigating the theoretical nature of oversmoothing. Oono & Suzuki (2020) performed an asymptotic analysis, showing that node embeddings homogenize when the number of layers tends to infinity. Wu et al. (2022) performed a non-asymptotic analysis, showing that oversmoothing occurs when an undesirable mixing effect overcomes a desirable denoising effect. Keriven (2022) showed that some smoothing *but not too much* can be desirable for linear GNNs, and that there exists a number of layers which optimizes this tradeoff. A major limitation of the existing body of work and an active area of research is the need of a formalized understanding of the relationship between homogenized node representations and model generalization. Many prior works such as DropEdge typically assume a clear relationship between oversmoothing reduction and model performance without formally justifying it. However, Keriven (2022) has made a step in this new direction, giving a theoretical analysis based on risk minimization, although it is limited to linear GNNs.

## 3 RANDOM DROPOUT AND OVERSMOOTHING

The authors of DropEdge Rong et al. (2020) and DropMessage Fang et al. (2023) propose to directly measure the amount of smoothing after applying each method. DropEdge (Rong et al., 2020) measures the difference in Euclidean distance between internal layers and the final layer. One criticism is that it does not distinguish between nodes in the same layer. In contrast, DropMessage (Fang et al., 2023) computes a metric over the nodes of the same layer, the mean average distance (MAD) (Chen et al., 2019). Notably, both methods are applied *exclusively during training*, which raises concerns regarding whether they can address oversmoothing at test time:

1. Generalization to unseen data: while these training-time interventions might help reduce oversmoothing on the training data, their absence during the test phase can potentially lead to inconsistent behaviour on the test data due to increased oversmoothing.

2. Model confidence: if the robustness against oversmoothing is only demonstrated during training and not during testing, it reduces the overall confidence in the model's reliability across diverse environments.

The distinction between training and testing time, and its implications on oversmoothing, remains unaddressed in the aforementioned works. This oversight has prompted our investigation into the effects of these random dropping methods on oversmoothing across both training and testing phases.

## 3.1 MEASURING SMOOTHING

One metric commonly used in the literature to empirically measure smoothing is *mean average distance* (MAD) (Chen et al., 2019):

$$d_{\mathrm{MAD}}(\mathbf{X}^\ell) = \frac{1}{|V|} \sum_{i \in V} \sum_{j \in \mathcal{N}_i} 1 - \frac{\mathbf{X}_i^{\ell\top} \mathbf{X}_j^\ell}{||\mathbf{X}_i^\ell \mathbf{X}_j^\ell||}. \tag{1}$$

More recently many works have proposed metrics of smoothing based on the concept of *Dirichlet energy* (Cai & Wang, 2020; Rusch et al., 2022) which is typically defined as

$$d_{\mathrm{DE}}(\mathbf{X}^\ell) = \frac{1}{|V|} \sum_{i \in V} \sum_{j \in \mathcal{N}_i} ||\mathbf{X}_i^\ell - \mathbf{X}_j^\ell||_2^2. \tag{2}$$

This has the property that $d_{\mathrm{DE}}(\mathbf{X}^\ell) = 0$ if and only if all node representations are equal – in other words, complete oversmoothing is equivalent to 0 Dirichlet energy, which has led to conceptually cleaner proofs (Cai & Wang, 2020) in the theoretical analysis of oversmoothing. However, note that the Dirichlet energy is sensitive to arbitrary scaling of embeddings. Observe that, by simply multiplying the embeddings by a constant greater than 1 after each layer, we are guaranteed to increase this energy. In reality, this might not reflect any improvement in the ability of the model to generalize. For our study, this may be problematic as DropMessage and standard dropout, which scale the embeddings – in the case of dropout with probability $p$ it is common to scale with $1/(1-p)$. This would 'fix' oversmoothing if a sufficiently high dropping probability is used compared to a model that applies less dropping. This is less of a concern when observing the layer-wise exponential convergence of embeddings within the same model.

Our experiments also aim to empirically compare the relative amount of oversmoothing suffered by different GNNs. As we specifically investigate methods that inherently scale the embeddings, we also consider using MAD, which has two known limitations: (i) complete oversmoothing (all node representations being identical) does not equate to 0 MAD, and (ii) it is ineffective in the case where node representations are scalars – nodes with the same sign but different magnitude cannot be distinguished. However, it can still be shown that for multidimensional MAD, there exist constants $C_1, C_2 > 0$ such that $\mu_{\mathrm{MAD}}(\mathbf{X}^\ell) \leq C_1 e^{-C_2 \ell}$ for $\ell \in [0, N]$— it exhibits layer-wise exponential convergence (Rusch et al., 2023). Although theoretically inconvenient, MAD still enables us to make meaningful empirical comparisons on oversmoothing on models where the node embeddings are high dimensional. Limitation (i) is not strictly a concern when we seek a comparison between methods, rather than an absolute measure of oversmoothing that is theoretically sound.

## 3.2 OBSERVING THE EFFECT OF RANDOM DROPPING ON OVERSMOOTHING

We empirically observe oversmoothing in a model by measuring the amount of smoothing after each layer. We compare a vanilla baseline with DropEdge and DropMessage by training models while applying random dropping and evaluate the extent of oversmoothing in two scenarios: (i) without applying any random dropping (simulating test-time inference), and (ii) with random dropping applied (resembling a training forward pass).

**Experimental setup.** Using 128-layer deep GNNs of the GCN architecture (Kipf & Welling, 2017), we train models on node classification tasks with varying levels of homophily. In addition to the commonly-used citation networks *Cora* (McCallum et al., 2000), *Citeseer* (Giles et al., 1998) and *Pubmed* (Sen et al., 2008), we use heterophilic datasets *Wisconsin*, *Texas*, *Cornell* (Craven et al., 1998) and *Chameleon* (Pei et al., 2020). As baselines, we use a vanilla model trained with skip connections (He et al., 2016), and also a model trained with dropout.

The results using Dirichlet energy are shown in Figure 1. We observe that at training time, DropEdge can alleviate the amount of smoothing at some layers by a scaling factor – however, the trend is still exponential: at layer $\ell$ the Dirichlet energy is $O(C^{-\ell})$ for some constant $C$. Layer-wise exponential convergence is still occurring. DropMessage, in contrast, is able to completely nullify it at training

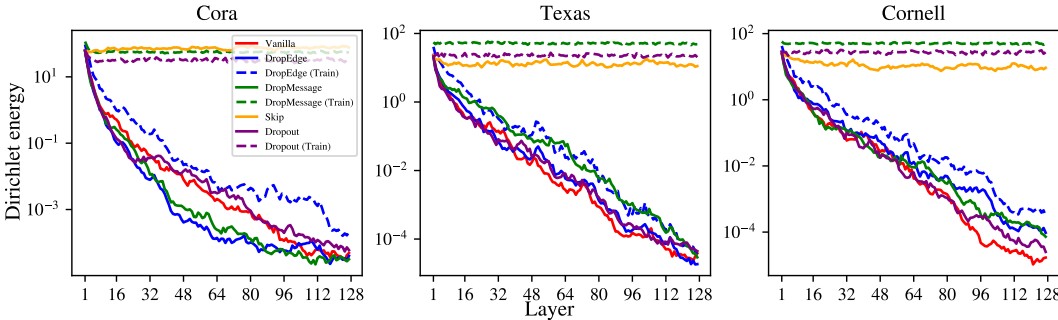

Figure 1: Measuring oversmoothing in random dropping models, averaged over 5 runs.

time, although it appears the same can also be achieved by applying dropout on the node vectors at each layer. Results using MAD are very similar, and given in A.3.

At test time, random dropping is not applied. Instead any improvement comes from the effect that the dropping has during training. However, we observe from our experiments (included in Table 1 in Section 5.2 for ease of later comparison) that naively enabling DropEdge and DropMessage at test time translates to poor accuracy and inconsistent model inference, *despite the reduction of oversmoothing*, suggesting that the main benefits of these random dropping methods is not primarily from oversmoothing reduction, or that oversmoothing reduction by itself is insufficient to guarantee improved performance.

**Discussion.**   During forward passes, DropEdge is able to mitigate the amount of oversmoothing, but does not appear to prevent it. The amount of mitigation is greater at training time than test time. DropMessage, in contrast, is able to stabilize the oversmoothing at training time, but has little effect at test time. If the primary cause of oversmoothing is the recursive aggregation inherent in the GNN's structure, this issue will still manifest at test time – it will aggregate information across all available edges without any dropping, which may lead to homogenized node representations.

We further note a close similarity between methods used outside the specific context of graphs. For instance, DropEdge is methodologically similar to *word dropout* (Mikolov et al., 2013) used in natural language processing and *cutout* (DeVries & Taylor, 2017) used in computer vision, both of which are augmentation techniques that aim to prevent the model from overfitting on a specific input feature. In addition, we note that DropMessage's approach is *effectively applying dropout between the aggregate and update stages of message passing*[1]. We observe that it has a similar effect on oversmoothing compared to applying dropout on the node representations.

If their primary effect was only through introducing noise during training, then the two methods would arguably be analogous to dropout and other similar approaches. Dropout aims to make neural networks more robust by preventing over-reliance on any particular neuron during training, but it does not drop out neurons at test time. Similarly, DropEdge/DropMessage can be seen as a way to ensure that the GNN does not over-rely on any particular edge or message.

In conclusion, DropEdge and DropMessage exhibit nuanced effects on smoothing at training time and are not applied at test time. They appear more aligned with robust training and overfitting prevention than directly combating the oversmoothing phenomenon. This suggests that these techniques, particularly DropEdge, might operate more as data augmentation. Their role in mitigating oversmoothing could then be an indirect outcome of the models they assist in training – models that are more robust and inherently resistant to oversmoothing. Further research is needed to explore the primary versus secondary effects of these techniques.

### 3.3 THE IMPORTANCE OF RANDOMNESS

An important implication of our results is the possibility that DropEdge actually does not have a significant effect in reducing oversmoothing. The overall behavior is still $O(k^{-L})$. This is contrary

---

[1]This can be verified using the source code: https://github.com/zjunet/DropMessage/blob/master/src/layer.py

to what is implied by the original work. We suspect that the reasons for DropEdge's performance improvements could lie elsewhere, and we investigate this.

We first recall that the authors of DropEdge use a specific definition of oversmoothing: they define the concept of $\epsilon$-*smoothing*, which occurs when all node representations lie within a distance $\epsilon$ from a subspace. It can be shown that during a forward inference pass, dropping edges can increase the layer at which (a relaxed version) of $\epsilon$-smoothing occurs. This is stated as Theorem 1 in the work by Rong et al. (2020), and we shall continue to refer to this theorem as the *DropEdge theorem*. The DropEdge theorem does not make any assumption on *how* edges are removed, it only requires that the number of edges in the perturbed graph be less than the original. Therefore, removing edges in a completely deterministic manner would also satisfy the theorem, but it is unclear whether this would lead to the same effects. This is the motivation for our next experiment.

**Experiment.**  We perform an investigation in which we train a model $\Phi$ using a version of DropEdge where a proportion of the edges, controlled by parameter $\tau \in [0, 1]$, are sampled deterministically. That is, we set a predefined set of edges $\mathcal{E} \subset E$ such that $|\mathcal{E}| = \lfloor p|E| \rfloor$. During the training of $\Phi$, at each epoch we choose the edges $\mathcal{F}$ to drop by sampling from $\mathcal{S} = \{\mathcal{E}' \mid |\mathcal{E}' \cap \mathcal{E}| \geq \lfloor \tau|E| \rfloor \wedge |\mathcal{E}'| = \lfloor p|E| \rfloor\}$. When $\tau = 0$, the method is equivalent to standard DropEdge. When $\tau = 1$, the method is a fully deterministic version of DropEdge where the same edges are sampled for all epochs. For different values of $\tau \in [0, 1]$ we observe the test accuracy of $\Phi$ and the MAD.

**Remark 3.1.** *$\tau$ controls the mutual information between $\mathcal{E}$ and $\mathcal{F}$*

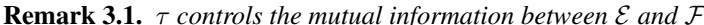

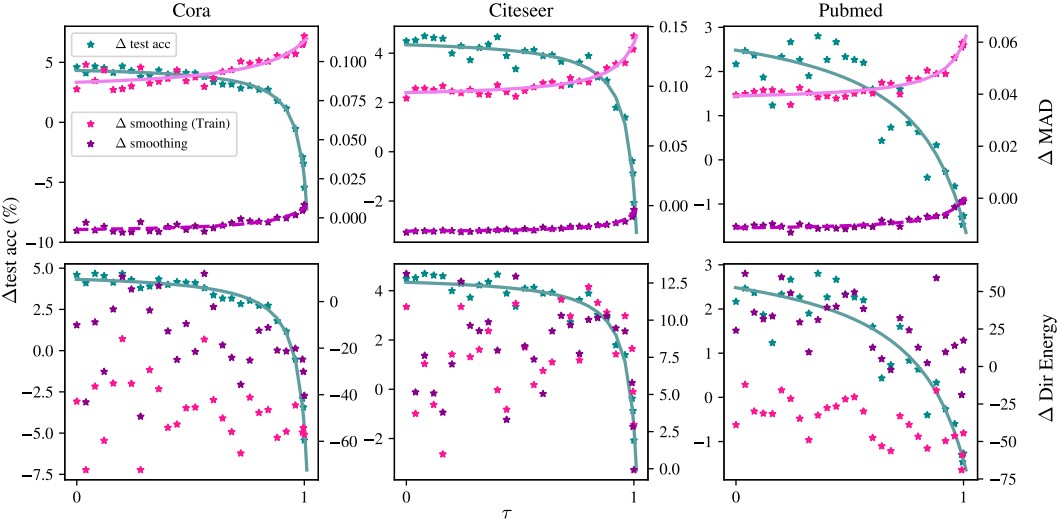

Figure 2: Investigating the effects of stochasticity in DropEdge.

In order to obtain stable results, we train shallower models with 3 GCN layers and skip connections. We repeat this experiment for multiple choices of the initial edge set $\mathcal{E}$ and measure the mean test accuracy and MAD at each $\tau$. Results using MAD and Dirichlet energy are shown in Figure 2. As $\tau$ increases, the performance improvement from using DropEdge degrades. Although satisfying the DropEdge theorem, for certain values of $\tau$ that are sufficiently large (typically after 0.9), the model has *worse* test performance than not using DropEdge, while the amount of oversmoothing surprisingly *decreases* according to MAD. Moreover, according to Dirichlet energy, there is *no consistent relationship* between performance improvement and smoothing. For Pubmed the Dirichlet energy tends to decrease as performance worsens, but for Citeseer there is an initial increase. This suggests:

- the performance improvement provided by DropEdge is highly sensitive to the amount of random noise being injected into the sampling during each training epoch.

- the amount of oversmoothing at both training and test time is possibly related to the model's error on the training/test sets, and how it generalizes (i.e. whether we find good minima).

As an additional check, we perform the same experiment but modifying the computation of MAD and Dirichlet energy so that only pairs of nodes with *different labels* are considered. The motivation is that smoothing across nodes with different labels is more problematic for performance, whereas smoothing across nodes with the same label is desirable. The resulting trends are very similar and shown in A.2 We can conclude that a GNN with reduced oversmoothing does not necessarily generalize better, which challenges the preconception made in previous works (Rong et al., 2020; Li et al., 2018; Chen et al., 2019) that less smoothing is strictly better.

## 4 LEARNING TO DROP

In Section 3 we highlighted several limitations of DropEdge and DropMessage. In summary, both DropEdge and DropMessage operate only during training, and do not sufficiently address over-smoothing on unseen data at test time. It is unclear whether oversmoothing is related to their effect on model performance. Moreover, dropout on the message matrix (as in the case of DropMessage) will stabilize both Dirichlet energy and MAD. However, doing this at test time will result in poor model performance and unstable predictions.

To address these concerns, we propose *learning to drop* (Learn2Drop) in which we *learn* which elements to drop rather than applying uniform treatment, and at test time choosing which elements to drop based on experience. This will incorporate domain knowledge into the dropping process, which may be advantageous over a pure adhoc approach where the dropping probability is fixed. This can (i) allow the model to keep essential information while filtering out noise, which has been previously observed to be a potential cause of oversmoothing (Chen et al., 2019), (ii) allow the use of topological information, which can affect smoothing (Bodnar et al., 2022), and (iii) perform dropping at test time in a more informed manner.

### 4.1 INFORMATION BOTTLENECK

Aligning with the motivation to preserve only critical information, we propose dropping messages based on the information bottleneck (IB) principle (Tishby et al., 2000). This principle has been previously adopted in neural networks for similar purposes, such as pruning less informative neurons (Achille & Soatto, 2018) and enhancing robustness against adversarial attacks (Kolchinsky et al., 2019). The overall idea is to seek a representation $Z$ that is minimally informative about the input $X$, whilst simultaneously being maximally informative about the target $Y$. This done by optimally balancing the mutual information terms $I(X, Z)$ and $I(Z, Y)$.

Recall that in message passing GNNs, at layer $\ell$ we obtain the next representation of node $i$, by applying an aggregation function $\oplus$ on the messages passed from the nodes in its neighborhood $\mathcal{N}_i$:

$$\mathbf{h}_i^{\ell+1} = \phi\left(\mathbf{h}_i^\ell, \oplus_{j \in \mathcal{N}_i}\left(\psi\left(\mathbf{h}_i^\ell, \mathbf{h}_j^\ell\right)\right)\right). \tag{3}$$

Here, $\psi\left(\mathbf{h}_i^\ell, \mathbf{h}_j^\ell\right)$ denotes the message that a node $j$ passes to a node $i$. Let $\mathbf{M}^\ell \in \mathbb{R}^{|E| \times m}$ be the message matrix given to layer $\ell$. It is formed by stacking all messages passed at layer $\ell$, and each row is a different message. Note that $\mathbf{M}^\ell$, along with the adjacency matrix of the input graph, are sufficient for obtaining the final output of the model. Thus, $\mathbf{M}^\ell$ can be viewed as some intermediate representation. In this view, the layers $1, \ldots, \ell-1$ of the message passing GNN can be treated as an encoder, and the remainder of the model can be viewed as a decoder. $\mathbf{M}^\ell$ contains the information necessary to make predictions about the target $Y$. This motivates us to apply the IB principle, treating $\mathbf{M}^\ell$ as the optimal representation $Z$. We shall refer to $\mathbf{M}^\ell$ as $Z$ for clarity.

Let $\phi$ be the parameters of the encoder and $\theta$ be the parameters of the decoder. We can write the mutual information between the input $X$ and the messages $Z$ as $I(X, Z; \phi)$, and the mutual information between the messages and the output $Y$ as $I(Z, Y; \theta)$. We can treat $Z$ as a random variable with distribution $\mathbb{P}(Z|X; \theta)$. Following standard use of the IB principle, we obtain

$$\max_{\theta, \phi} I(Z, Y; \theta) - \beta I(X, Z; \phi). \tag{4}$$

Optimizing this objective will allow us to obtain minimal and sufficient representations of the input graph. This objective is intractable. Following (Alemi et al., 2017; Wu et al., 2020; Miao et al.,

2022), we use variational approximations: $\mathbb{P}_\phi$ to approximate the encoder, and $\mathbb{Q}_\theta$ to approximate the decoder, and $\mathbb{R}(Z)$ to approximate the marginal distribution of $Z$. This yields the variational bounds:

$$I(X, Z; \phi) \leq \mathbb{E}_{X,Z} \text{KL}(\mathbb{P}_\phi(Z|X) \,||\, \mathbb{R}(Z)) \tag{5}$$

$$I(Z, Y; \theta) \geq \mathbb{E}_{Z,Y}[\log \mathbb{Q}_\theta(Y \mid Z)] + H(Y) \tag{6}$$

It can be shown using the standard derivation introduced by Alemi et al. (2017) that applying these variational bounds results in the objective

$$\max_{\theta,\phi} \mathbb{E}[\log \mathbb{Q}_\theta(Y \mid Z)] - \beta \mathbb{E}[\text{KL}(\mathbb{P}_\phi(Z|X) \,||\, \mathbb{R}(Z))]. \tag{7}$$

Here, the first term is the expected negative log-likelihood. For classification tasks this is equivalent to the cross entropy loss. The second term is harder to evaluate, and depends on the instantiation of the encoder $\mathbb{P}_\phi$.

## 4.2 Instantiating the distributions

We have a choice of distribution for the encoding of $Z$. Recall that our motivation is to optimize the dropping of information from the messages. We can do this probabilistically. Specifically, we obtain each element of every message by sampling from a *spike and slab* distribution (Ishwaran & Rao, 2005). Each distribution is parameterized by a value $v$ and a sample probability $p$. The value $v$ is sampled with probability $p$, and $0$ is sampled with probability $1-p$. In the context of our method, we consider each message element as a variable that can be either retained or discarded. This captures the notion of allowing elements to be dropped. We choose the slab as a Delta function $\delta(x - l)$. The parameters $l$ and $p$ for each distribution are learned during training.

Thus, sampling from $\mathbb{P}_\phi$ is equivalent to sampling from a set of spike-and-slab distributions, where each distribution is parameterized by a different value in the original message matrix (prior to dropping). In practice, for a message vector $\mathbf{q}_{ij}$ – the message passed from node $j$ to $i$ – we can obtain the vector of probabilities by feeding the concatenated node representations $[\mathbf{h}_i^{\ell-1}||\mathbf{h}_j^{\ell-1}]$ into an MLP. The final message vector after dropping can be obtained using the Gumbel Sigmoid trick (Jang et al., 2017) to allow the gradients to flow through the learned probabilities. Doing this for all messages will compute the optimized representation $Z$.

We can define $\mathbb{R}(Z)$, the variational approximation of the marginal $P(Z)$, by sampling each message element $q_{ij}^k$ from set of spike and slab distributions sharing the same spike probability $r \in [0, 1]$, as well as the same uniform slab distribution $\text{Uniform}(a, b)$ where $a, b \in \mathbb{R}$. This gives $\mathbb{R}(Z) = \prod P(q_{ij}^k)$ where $P(q_{ij}^k) = \mathbb{P}(x) = \frac{p}{b-a} + (1-p)\delta(x)$.

Now, computing the KL term in Equation 7 directly is intractable, as it requires a summation over all possible $Z$. Instead, we note that since the elements of $Z$ are independent given $X$, the joint distribution $\mathbb{P}(Z \mid X)$ can be factorized into the product of the individual marginal distributions of each element of $Z$. That is, $\prod P(v_{ij}^k \mid X)$, where $v_{ij}^k$ refers to the $k$-th message element of the message passed from $j$ to $i$. The KL divergence then has the analytical form

$$\sum_{(ij) \in E, k \in [m]} -(1 - p_{ij}^k) \log \frac{1 - p_{ij}^k}{1 - r} - \int_{-\infty}^{\infty} p_{ij}^k \delta(x - l_{ij}^k) \log \frac{p_{ij}^k \delta(x - l_{ij}^k)}{1/(b-a)} \, dx$$

$$= \sum_{(ij) \in E, k \in [m]} -(1 - p_{ij}^k) \log \frac{1 - p_{ij}^k}{1 - r} - p_{ij}^k \log \frac{p_{ij}^k(b-a)}{r}, \tag{8}$$

where $p_{ij}^k$ and $l_{ij}^k$ are the spike-and-slab parameters for each message element. This is sum of the KL divergences of the marginal distributions of each message element.

## 5 Experiments on oversmoothing

In this section, we evaluate the effectiveness of Learn2Drop. We first show that Learn2Drop is able to successfully mitigate oversmoothing at test time, whereas DropEdge and DropMessage are unable to. We then evaluate the performance of Learn2Drop against previous dropping approaches to investigate whether it helps model performance in practice.

## 5.1 OVERSMOOTHING REDUCTION

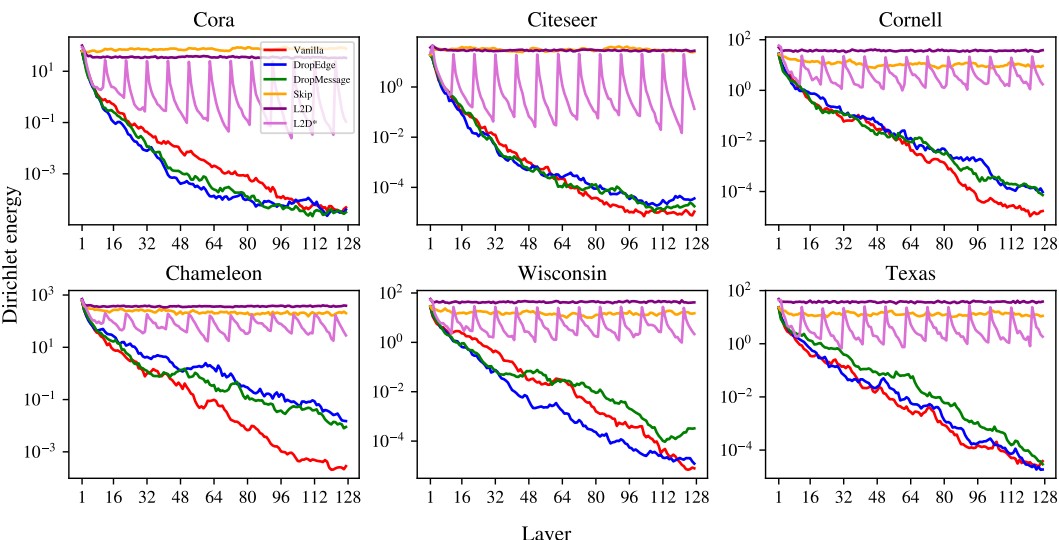

Figure 3: Oversmoothing comparison across six node classification datasets.

Using the same methodology as in 3.2 we measure the amount of test-time oversmoothing in models trained using Learn2Drop. We evaluate two versions of Learn2Drop: one where dropping is performed at every layer (denoted L2D) and another where dropping is only performed once every ten layers (L2D*), since dropping at every layer for very deep GNNs may add unnecessary overhead. The results using Dirichlet energy are shown in Figure 3, and corresponding results using MAD are given in Appendix A.3. We observe that for each task, Learn2Drop results in a *significant reduction* in smoothing according to both metrics. Interestingly, from observing L2D*, we observe that applying a single dropping layer is able to reset the Dirichlet energy. While for DropEdge and DropMessage, there is an increase in oversmoothing at a super-linear rate, it is clear that Learn2Drop keeps oversmoothing from changing beyond one order of magnitude.

## 5.2 MODEL PERFORMANCE

In prior work, it has been standard to perform an indirect evaluation on oversmoothing by training very deep GNNs and showing that the usual performance degradation (compared to a shallow model) is reduced (Rong et al., 2020). For instance, whereas a vanilla GCN would suffer a significant reduction in performance on Cora if we were to use 64 layers instead of the usual 2 to 3 that typically yields optimal performance, DropEdge may only suffer a moderate hit. However, as discussed in Section 3.3, the effects of overfitting and oversmoothing are likely interlaced and difficult to decouple. It is not evident from such observations whether the model is simply more resistant against overfitting, or whether oversmoothing is actually reduced, especially since one of these may be the indirect consequence of the other. Nevertheless, it may be beneficial to observe the performance of models in such scenarios where a combination of issues is prevalent.

For each dataset, we train 3-layer, 32-layer, and 64-layer GCN models. We compare the test time accuracy against both default and test-time enabled versions of DropEdge and DropMessage, as well as an additional baseline where dropout with probability $0.5$ is applied after each layer. Moreover, in the context of evaluating oversmoothing, which purportedly occurs at deeper layers, we desire a scenario where it is beneficial to use more layers. Following Zhao & Akoglu (2020) we opt for using a 'missing feature' setting where 90% of the nodes have their feature vector initialized to $\mathbf{0}$.

The results shown in Table 1. Learn2Drop is able to successfully mitigate the performance degradation when increasing the number of layers. Note that the test-time versions of DropEdge and DropMessage (denoted with ∗), despite reducing oversmoothing, perform highly inconsistently resulting in poor performance, and often fail to converge. To make this baseline more sensible, we obtain each individual result by averaging 10 forward passes. Meanwhile, Learn2Drop consistently

achieves higher test accuracy, perhaps as it *learns* the optimal way to drop. One can view this as a mechanism that controls the amount of smoothing reduction by using the IB principle to optimally make the tradeoff between signal and noise. However, we emphasize that this is merely a hypothesis. On the contrary, it could be that the oversmoothing reduction is a side effect of the true mechanisms underlying the performance improvement, which is what we have examined for DropEdge.

Here we focus on understanding the effect of various random dropping techniques on model performance. Competing with the state-of-the-art techniques that address oversmoothing is not the objective. For completeness, we have included the recent method GraphCON (Rusch et al., 2022) which has specifically been designed to combat oversmoothing and outperforms all dropping approaches.

| L | | Cora | Citeseer | Cornell | Chameleon | Wisconsin | Texas |
|---|---|---|---|---|---|---|---|
| **3** | **Vanilla** | $64.2 \pm 0.7$ | $44.0 \pm 1.1$ | $45.4 \pm 7.3$ | $28.4 \pm 1.2$ | $46.3 \pm 9.3$ | $56.2 \pm 6.7$ |
| | **DropEdge** | $66.0 \pm 2.4$ | $44.5 \pm 1.4$ | $44.3 \pm 5.6$ | $27.5 \pm 2.5$ | $46.3 \pm 8.3$ | $\mathbf{57.3 \pm 5.5}$ |
| | **Dropout** | $65.1 \pm 3.3$ | $46.2 \pm 2.4$ | $43.8 \pm 5.2$ | $29.3 \pm 2.6$ | $45.9 \pm 8.9$ | $55.7 \pm 4.7$ |
| | **DropMessage** | $64.4 \pm 2.4$ | $48.0 \pm 2.0$ | $47.5 \pm 5.3$ | $27.9 \pm 2.8$ | $\mathbf{51.0 \pm 4.6}$ | $56.6 \pm 4.4$ |
| | **L2D** | $\mathbf{66.4 \pm 1.3}$ | $\mathbf{49.1 \pm 2.5}$ | $\mathbf{48.5 \pm 4.7}$ | $\mathbf{30.2 \pm 3.3}$ | $51.0 \pm 2.4$ | $56.4 \pm 4.3$ |
| | *DropEdge\** | *$58.9 \pm 4.3$* | *$42.3 \pm 1.3$* | *$42.2 \pm 4.9$* | *$25.7 \pm 2.9$* | *$44.3 \pm 6.4$* | *$55.7 \pm 5.3$* |
| | *DropMessage\** | *$60.3 \pm 3.0$* | *$43.0 \pm 1.3$* | *$40.5 \pm 1.9$* | *$23.6 \pm 3.5$* | *$47.1 \pm 4.6$* | *$37.3 \pm 21$* |
| | *GraphCon* | *$68.5 \pm 3.2$* | *$52.1 \pm 0.9$* | *$53.1 \pm 2.3$* | *$35.2 \pm 5.3$* | *$53.4 \pm 2.4$* | *$58.5 \pm 3.1$* |
| **32** | **Vanilla** | $70.5 \pm 1.5$ | $51.2 \pm 1.4$ | $44.3 \pm 5.6$ | $27.6 \pm 2.6$ | $49.4 \pm 7.6$ | $58.9 \pm 5.5$ |
| | **DropEdge** | $68.6 \pm 1.7$ | $47.9 \pm 1.7$ | $40.0 \pm 5.0$ | $30.0 \pm 2.2$ | $46.3 \pm 8.0$ | $57.8 \pm 5.3$ |
| | **Dropout** | $23.6 \pm 7.6$ | $22.3 \pm 3.3$ | $44.3 \pm 5.6$ | $20.4 \pm 1.9$ | $48.6 \pm 8.1$ | $57.8 \pm 5.8$ |
| | **DropMessage** | $66.2 \pm 1.7$ | $50.5 \pm 1.4$ | $47.1 \pm 7.2$ | $28.3 \pm 2.1$ | $50.0 \pm 4.5$ | $57.3 \pm 4.5$ |
| | **L2D** | $\mathbf{72.4 \pm 2.5}$ | $\mathbf{52.3 \pm 4.6}$ | $\mathbf{47.4 \pm 6.4}$ | $\mathbf{30.5 \pm 2.1}$ | $\mathbf{51.0 \pm 2.7}$ | $\mathbf{59.8 \pm 2.4}$ |
| | *DropEdge\** | *$62.9 \pm 3.7$* | *$44.8 \pm 2.8$* | *$44.3 \pm 6.2$* | *$27.9 \pm 1.7$* | *$40.8 \pm 5.3$* | *$58.4 \pm 6.2$* |
| | *DropMessage\** | *$53.2 \pm 20$* | *$29.4 \pm 3.2$* | *$16.8 \pm 3.0$* | *$16.4 \pm 3.3$* | *$21.6 \pm 11$* | *$14.1 \pm 6.5$* |
| | *GraphCon* | *$76.6 \pm 4.6$* | *$53.1 \pm 1.2$* | *$49.3 \pm 3.2$* | *$33.4 \pm 4.3$* | *$52.3 \pm 3.4$* | *$59.6 \pm 3.1$* |
| **64** | **Vanilla** | $47.2 \pm 14.3$ | $48.3 \pm 3.4$ | $44.3 \pm 5.6$ | $27.6 \pm 1.7$ | $46.3 \pm 7.2$ | $58.4 \pm 5.6$ |
| | **DropEdge** | $66.4 \pm 4.0$ | $45.7 \pm 0.7$ | $43.2 \pm 5.9$ | $27.4 \pm 0.8$ | $42.7 \pm 3.1$ | $55.7 \pm 7.2$ |
| | **Dropout** | $18.3 \pm 7.0$ | $21.8 \pm 2.4$ | $44.3 \pm 5.6$ | $21.4 \pm 1.4$ | $47.8 \pm 8.8$ | $58.4 \pm 5.6$ |
| | **DropMessage** | $65.2 \pm 2.1$ | $48.6 \pm 1.5$ | $45.8 \pm 7.5$ | $\mathbf{28.9 \pm 2.6}$ | $\mathbf{50.0 \pm 3.3}$ | $55.9 \pm 5.6$ |
| | **L2D** | $\mathbf{69.3 \pm 3.6}$ | $\mathbf{50.5 \pm 2.6}$ | $\mathbf{47.3 \pm 4.7}$ | $28.7 \pm 1.4$ | $49.0 \pm 1.7$ | $\mathbf{59.3 \pm 1.5}$ |
| | *DropEdge\** | *$29.7 \pm 5.0$* | *$29.0 \pm 4.9$* | *$44.3 \pm 6.2$* | *$27.0 \pm 0.7$* | *$47.1 \pm 8.0$* | *$56.2 \pm 5.2$* |
| | *DropMessage\** | *$16.7 \pm 7.1$* | *$23.8 \pm 6.1$* | *$18.9 \pm 9.2$* | *$17.3 \pm 1.9$* | *$20.8 \pm 10$* | *$20.0 \pm 21$* |
| | *GraphCon* | *$71.4 \pm 3.4$* | *$53.3 \pm 1.2$* | *$50.5 \pm 2.2$* | *$33.6 \pm 3.7$* | *$56.3 \pm 3.6$* | *$59.6 \pm 5.6$* |

Table 1: Comparison of test accuracy for different models and datasets with different backbone models, averaged over 5 runs. The highest accuracy across random dropping approaches is boldened.

## 6    Conclusion

In summary, we investigate the relationship between random dropping approaches and their ability to reduce oversmoothing. Specifically, while DropEdge introduces a degree of robustness, its direct impact on addressing oversmoothing at test time appears limited. We hypothesize that its effects are similar to data augmentation and support this with empirical results. DropMessage has a more pronounced effect but is still a training phase technique. In response to the the difficulty in directly applying dropping methods at test time, we present Learn2Drop, which decides which parts of the message matrix to keep or discard based on the information's relevance. This approach allows us to leverage the effect of dropping at test time in a more informed manner. Learn2Drop, like many previous methods, reduce oversmoothing while improving performance. However, this is not a guarantee that the performance improvement is a consequence of this oversmoothing reduction.

Our work provides novel emprical results that align with (Keriven, 2022)'s theoretical analysis, suggesting that always seeking to minimize oversmoothing is not optimal. An important takeaway from our work is that in practice, oversmoothing reduction will not strictly boost a GNN's performance – it is trivial to minimize oversmoothing by dropping messages randomly at test time. We have observed that a GNN with little oversmoothing does not guarantee optimal performance, and a GNN that experiences less oversmoothing is not necessarily more accurate. There is perhaps a general misconception that reducing oversmoothing is always desirable because many of the earlier methods proposed to tackle oversmoothing also implicitly introduce some form of regularization. We hope that future research can shed more clarity on this.

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

# A    ADDITIONAL RESULTS

## A.1    SMOOTHING OF DROPEDGE AND DROPMESSAGE

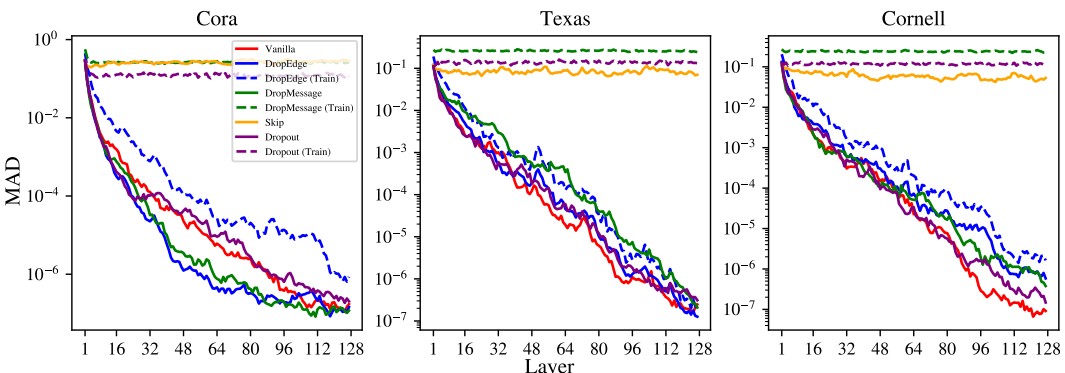

Figure 4: Full oversmoothing comparison using mean average distance.

## A.2    DROPEDGE OVERSMOOTHING VS TEST ACCURACY

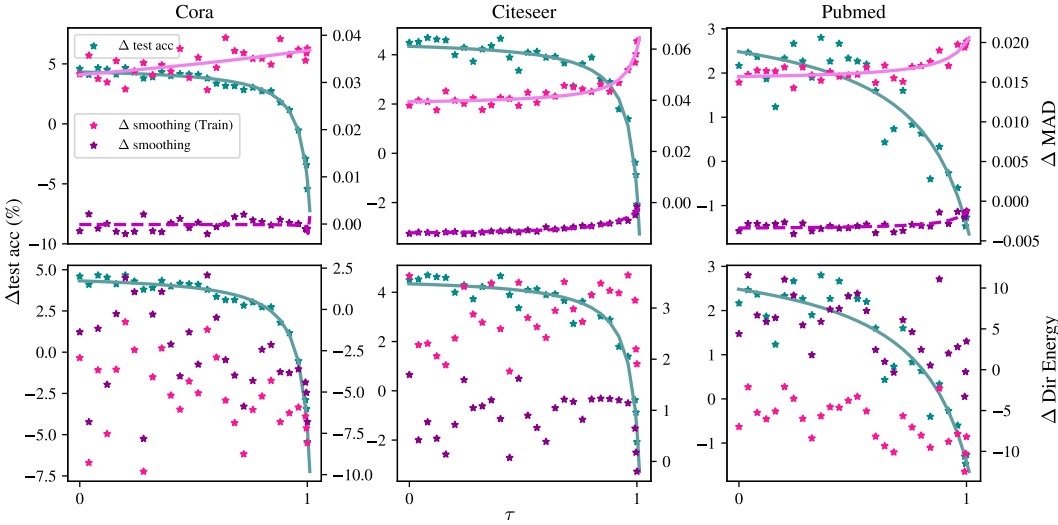

Figure 5: Oversmoothing and test accuracy relationship when DropEdge is applied. Oversmoothing here is calculated on only node pairs with different labels.

## A.3 LEARN2DROP OVERSMOOTHING

Figure 6: Oversmoothing comparison across six node classification datasets, using MAD.

## B  EXPERIMENTAL SETUP

We provide details of the model architectures and parameter settings for each of our experiments. All of our experiments were conducted using a V100 GPU on a Colab Pro+ subscription.

### B.1  EXPERIMENTS IN SECTION 3.3 (OVERSMOOTHING VS MODEL PERFORMANCE)

For this experiment, we train shallow 3-layer GNNs on Cora, Citeseer and Pubmed using the GCN architecture provided by `torch-geometric`[2]. The Adam optimizer from `pytorch` is used for all experiments in this work. The train-validation-test splits are the ones defined by setting the `public` parameter in `torch-geometric`. The DropEdge probability is fixed at 0.5.

| Parameter | Cora | Citeseer | Pubmed |
|---|---|---|---|
| Learning Rate | 0.003 | 0.005 | 0.003 |
| Training Epochs | 2000 | 2000 | 2000 |
| Early Stopping | 150 | 150 | 150 |
| Embedding Size | 32 | 32 | 32 |
| Number of Layers | 3 | 3 | 3 |

Table 2: Training and Implementation Details for GNNs on Different Datasets

### B.2  EXPERIMENTS IN SECTION 5.1 (OVERSMOOTHING ANALYSIS)

In this section we mainly focus on observing oversmoothing for individual models rather than strictly comparing them. There is no rigourous grid search of parameters. The same settings are used across all methods for the same dataset. $0.5$ is used as the dropping probability for both DropMessage and DropEdge.

---

[2]https://pytorch-geometric.readthedocs.io/en/latest/

| Parameter | Cora | Citeseer | Cornell | Chameleon | Wisconsin | Texas |
|---|---|---|---|---|---|---|
| Learning Rate | 0.01 | 0.005 | 0.002 | 3e-5 | 2e-5 | 5e-5 |
| Training Epochs | 2000 | 2000 | 2000 | 3500 | 500 | 500 |
| Early Stopping | 150 | 150 | 150 | 450 | 150 | 150 |
| Embedding Size | 32 | 32 | 32 | 32 | 32 | 32 |
| Number of Layers | 128 | 128 | 128 | 128 | 128 | 128 |

Table 3: Training details for GNNs

## B.3 EXPERIMENTS IN SECTION 5.2 (MODEL PERFORMANCE)

In these experiments we attempt to compare model performance across different randrom dropping methods. As such, we perform a grid search on the learning rate. The ranges of this search are specified in the table and the search space is logarithmic. The same settings are used across all methods for the same dataset.

| Parameter | Cora 3 | Citeseer 3 | Cornell 3 | Chameleon 3 | Wisconsin 3 | Texas 3 |
|---|---|---|---|---|---|---|
| Learning Rate | 2e-4 to 1e-6 | 2e-4 to 1e-6 | 2e-4 to 1e-6 | 1e-5 to 1e-6 | 1e-5 to 1e-6 | 2e-5 to 2e-6 |
| Training Epochs | 2000 | 2000 | 2000 | 3500 | 500 | 500 |
| Early Stopping | 150 | 150 | 150 | 450 | 150 | 150 |
| Embedding Size | 32 | 32 | 32 | 32 | 32 | 32 |

Table 4: 3 layer models

| Parameter | Cora 32 | Citeseer 32 | Cornell 32 | Chameleon 32 | Wisconsin 32 | Texas 32 |
|---|---|---|---|---|---|---|
| Learning Rate | 2e-4 to 1e-6 | 2e-4 to 1e-6 | 2e-4 to 1e-6 | 1e-5 to 1e-6 | 1e-5 to 1e-6 | 2e-5 to 2e-6 |
| Training Epochs | 4000 | 4000 | 4000 | 4500 | 5500 | 5500 |
| Early Stopping | 550 | 550 | 550 | 1450 | 1450 | 1450 |
| Embedding Size | 32 | 32 | 32 | 32 | 32 | 32 |

Table 5: 32 layer models

| Parameter | Cora 64 | Citeseer 64 | Cornell 64 | Chameleon 64 | Wisconsin 64 | Texas 64 |
|---|---|---|---|---|---|---|
| Learning Rate | 1e-3 to 2e-5 | 1e-3 to 2e-5 | 2e-4 to 2e-5 | 9e-4 to 9e-6 | 9e-4 to 9e-6 | 9e-4 to 9e-6 |
| Training Epochs | 4000 | 4000 | 4000 | 4500 | 5500 | 5500 |
| Early Stopping | 550 | 550 | 550 | 1450 | 1450 | 1450 |
| Embedding Size | 32 | 32 | 32 | 32 | 32 | 32 |

Table 6: 64 layer models

Moreover, there are method-specific parameters obtained using grid search.

| Parameter | Start | End | Search Space |
|---|---|---|---|
| DropEdge $p$ | 0.2 | 0.8 | Linear |
| DropMessage $p$ | 0.2 | 0.8 | Linear |
| L2D $\beta$ | 0 | 100 | Logarithmic |

Table 7: Parameter Search Ranges

