# A ADDITIONAL RESULTS

## A.1 OVERSMOOTHING IN DROPEDGE AND DROPMESSAGE

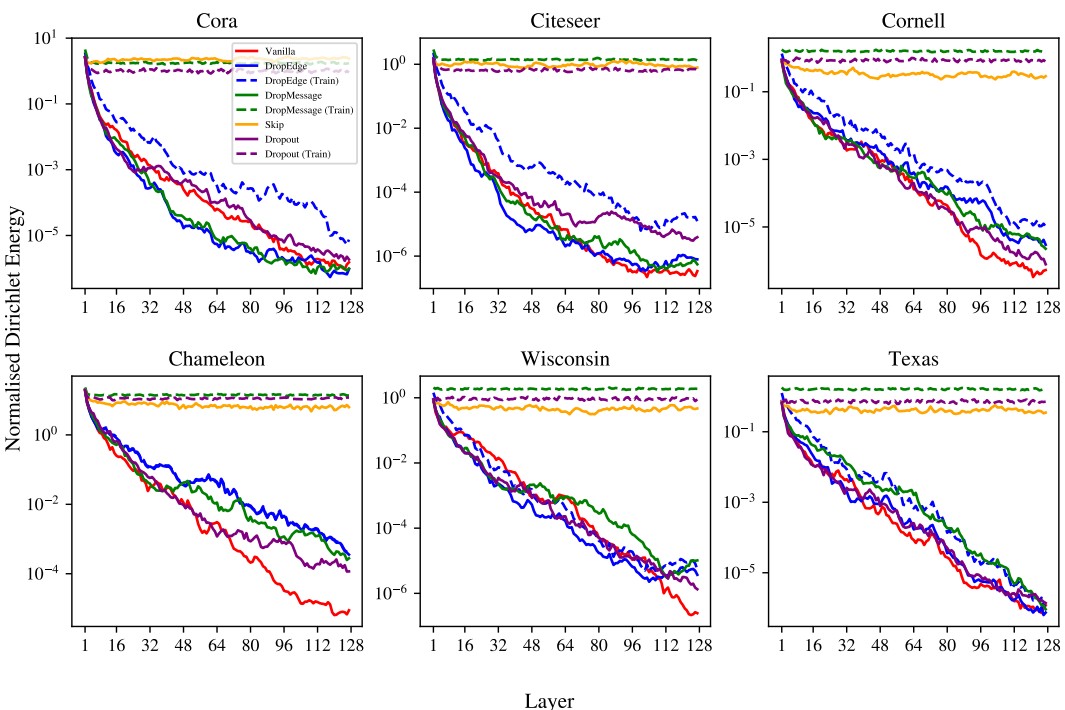

Figure 4: Full oversmoothing comparison using normalized Dirichlet energy.

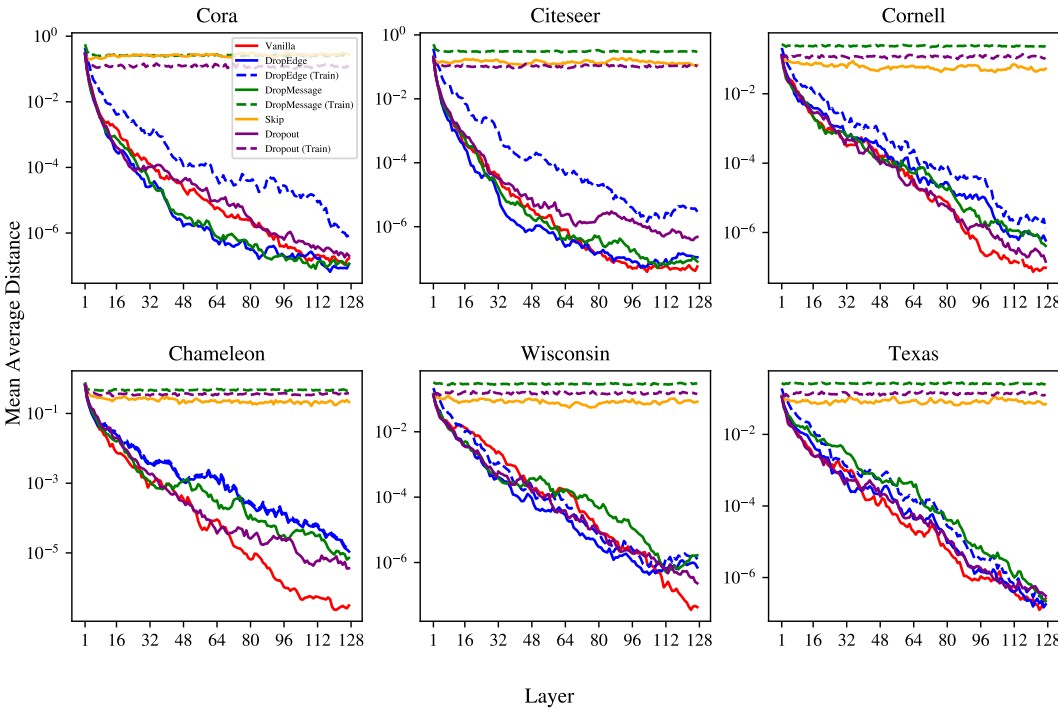

Figure 5: Full oversmoothing comparison using mean average distance.

## A.2 LEARN2DROP OVERSMOOTHING

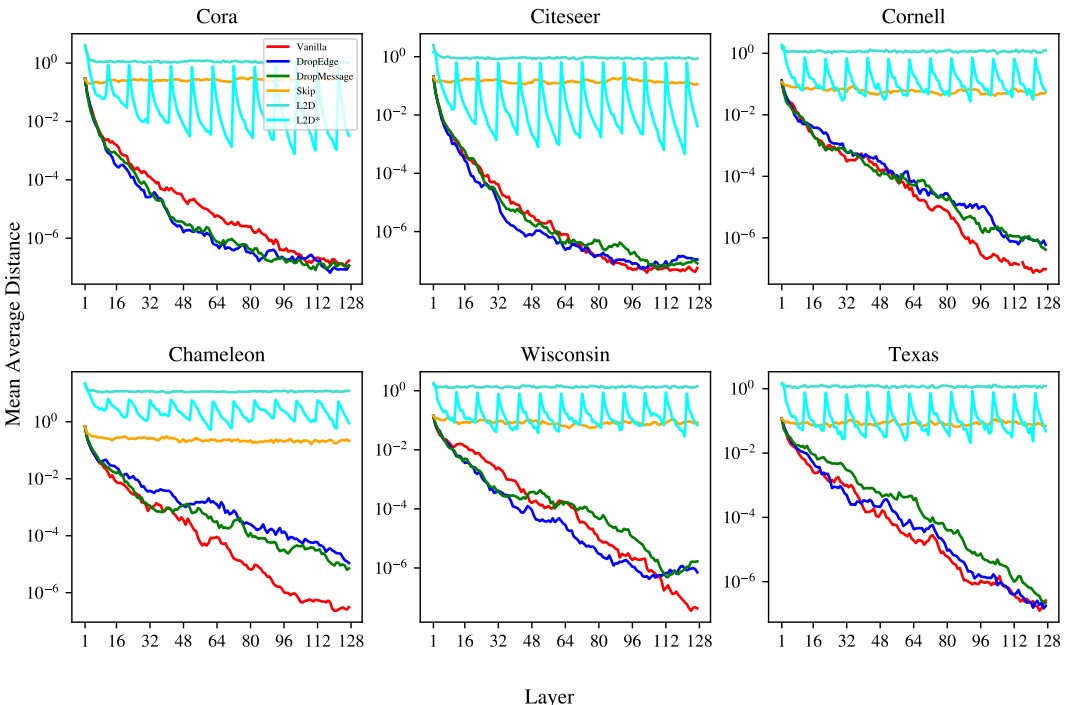

Figure 6: Oversmoothing comparison across six node classification datasets, using MAD.