# OpenReview forum: "On information dropping and oversmoothing in graph neural networks"
_ICLR.cc/2024/Conference — Submitted to ICLR 2024_

### Official Review · Reviewer_yTML · 2023-10-20

**Soundness:** 2 fair
**Presentation:** 2 fair
**Contribution:** 2 fair
**Rating:** 3
**Confidence:** 3

**Summary:**

The paper delves into a crucial issue within Graph Machine Learning, namely, the challenge of excessive smoothing in graph data. The authors aim to tackle this problem by applying the information bottleneck principle and demonstrate its effectiveness through the introduction of a new metric they've coined, the "normalized Dirichlet energy." In their research, the authors illustrate the advantages of their method, as it succeeds in reducing the Dirichlet energy while also yielding some improvements in classification accuracy.

**Strengths:**

The paper outlines a novel application of the information bottleneck principle for mitigating oversmoothing in graph data. Additionally, the paper endeavors to demonstrate that there is no discernible correlation between oversmoothing and test performance, as suggested by their research findings.

**Weaknesses:**

Please refer to questions.

**Questions:**

I encountered challenges in comprehending certain sections of the paper, particularly sections 4.1 and 4.2. I believe the clarity of these sections could be significantly improved with revisions by the authors. Additionally, I have a few follow-up questions:

1. Although the authors correctly emphasize the absence of a strong correlation between oversmoothing and test performance, it is essential to explore the specific conditions under which this correlation holds. This inquiry is vital because scenarios exist where the Normalized Dirichlet Energy may approach zero, yet the test accuracy remains high. For instance, consider a stochastic block matrix with two classes, where all nodes within one class map into a single representation, and the same occurs for the other class (to a different single representation. In this case, the Normalized Dirichlet Energy would be small, but test accuracy would be high. This raises questions about the circumstances in which oversmoothing genuinely matters and whether the metric employed is appropriate.

2. Building upon the first point, the connection between oversmoothing and generalization performance appears to be meaningful primarily when oversmoothing occurs across different classes. Oversmoothing within the same class might not be as relevant. Therefore, it would be valuable for the authors to provide commentary, theory, and experimental evidence in this context, while also demonstrating the efficacy of their method in comparison to other approaches.

3. My concerns extend to the topic of reproducibility. I was unable to locate any provided code or information regarding hyperparameter ranges. Inclusion of this information is crucial for transparency and for ensuring that other researchers can replicate the experiments.

In conclusion, the authors do illustrate that their method reduces the Normalized Dirichlet Energy. However, the actual benefits of their method for node classification tasks remain unclear, as well as its performance when applied to more complex graph architectures. Furthermore, it's uncertain whether their method is effective in the context of homophilic or heterophilic graphs. Therefore, based on these questions and concerns, I am inclined to recommend rejecting the paper. However, I am open to reconsidering this recommendation if the authors provide satisfactory responses and address these issues adequately.

---

> ### Author Response · Authors · 2023-11-17
> **Response to Reviewer yTML**
>
> Dear Reviewer yTML
>
> Thank you for your response and important feedback! We agree with you and realize that the original draft of our paper had major clarity issues, especially in sections 4.1 and 4.2, and the overall motivations and conclusions of the paper were not clearly explained. **We have revised large portions of the paper to address the clarity issues**.
>
> > I encountered challenges in comprehending certain sections of the paper, particularly sections 4.1 and 4.2. I believe the clarity of these sections could be significantly improved
>
> We have made Section 4 clearer to read – in particular we have corrected notation that was inconsistent and convoluted (especially regarding the optimized representation / message matrix). The derivation there is standard in the literature and is not the main contribution of our work. We have made clearer references to the foundational works on the IB principle in neural networks. We hope this section is now more understandable.
>
> The introduction, literature review and conclusion sections have been revised to make it clearer what the paper’s objectives and takeaways are. To clarify, we focus on investigating the effect of random dropping approaches on oversmoothing, and the degree to which oversmoothing is relevant to improving model performance in practice. Our proposal of Learn2Drop mainly serves to address the problems of using existing random dropping approaches at test time.
>
> > scenarios exist where the Normalized Dirichlet Energy may approach zero, yet the test accuracy remains high.
>
> Your concerns on the use of Normalized Dirichlet energy (NDE) are valid and it is true that we did not motivate its use properly. Reviewer K9Rv had similar concerns.  **As such we have completely removed the use of normalized Dirichlet energy from the paper and have redone the experiments using both standard Dirichlet energy and MAD**. We added a discussion on the benefits and pitfalls of using either metric in Section 3.1. The empirical results after switching to standard Dirichlet energy and MAD are very similar to our original results, and our conclusions remain the same. This suggests that the theoretical differences between each metric are not significant in practice, at least for our investigation.
>
>  > Oversmoothing within the same class might not be as relevant
>
> This is an interesting take and suggestion. The existing metrics of Dirichlet energy and MAD look indiscriminately at all node pairs (which are connected by an edge). It would be interesting to observe oversmoothing for only the pairs of nodes that are of different classes, since differentiating between nodes is what matters for classification accuracy. We did some additional experiments (for section 3.2) where we compute Dirichlet energy and MAD while only considering these pairs of nodes in different classes. The plots for those are placed in the Appendix. The overall results are highly similar to when comparing all pairs, although the absolute value of the smoothing differs. However, we are mostly concerned with the trend of this metric, rather than its absolute value.
>
> > My concerns extend to the topic of reproducibility.
>
> We have tried to improve the reproducibility of our work by including model training and parameter selection details in a new experimental setup section in the Appendix.
>
> We hope our revisions to the paper have addressed your concerns and would be grateful for any further comments or feedback.
>
> Thank you,
>
> The authors.

---

> > ### Comment · Reviewer_yTML · 2023-11-19
> >
> > I've reviewed both the paper and its revised version, and I've identified a few comments and questions.
> >
> > 1. Upon examining the paper, it is evident that there have been substantial changes, transitioning from presenting a novel method to focusing on benchmarking and understanding oversmoothing. Given this shift, wouldn't it be prudent to take a step back and refine the conceptual framework and  resubmit to the next conference? Considering the considerable alteration in messaging, this approach might be more judicious.
> >
> > 2. I find myself somewhat uncertain about grasping the paper's contribution fully. Comparing it to the paper: GraphCon by Rusch (2022), which had some theoretical contributions (albeit not highly rigorous), and noting that the current version also makes similar contributions of a somewhat theoretical nature, I'm still grappling with a comprehensive understanding. At a broad level, it seems that the paper aims to raise inquiries about scenarios where reducing oversmoothing may not necessarily lead to performance improvements. However, this conclusion appears somewhat obvious, especially when considering the illustrative example I presented in my previous comment. Consequently, I remain unclear about how or why this paper would bring tangible benefits to the broader community.
> >
> > It would help if the authors could respond to the above two comments.

---

> > > ### Author Response · Authors · 2023-11-21
> > > **Our response**
> > >
> > > Dear Reviewer yTML,
> > >
> > > Thank you for your continued engagement with our paper and for sharing your insights. We appreciate your suggestion about refining our conceptual framework and acknowledge the changes in our paper's focus from the original version. However, we believe that the current version of the paper makes an important empirical contribution to the field and merits consideration in its present form.
> > >
> > > **Empirical Contribution**: One of our key contributions is the empirical demonstration that commonly used random dropping approaches, such as DropEdge and DropMessage, do not effectively reduce oversmoothing as previously believed. This finding *challenges widely accepted notions* in the field, as evidenced by the extensive citation of DropEdge and the recognition of DropMessage as a distinguished paper at AAAI this year [1]. Our work thus serves to *correct a misconception* in current literature, providing a crucial perspective that is necessary for the community to consider.
> > >
> > > **Beyond Theoretical and Illustrative Examples**: While we agree that counterexamples to the notion that 'less oversmoothing is better' may seem trivial, our work goes beyond these simple illustrations. We address the gap in the literature where most authors do not empirically examine the relevance of oversmoothing in depth. Our rigorous experimentation contributes to a more nuanced understanding of oversmoothing, moving beyond theoretical constraints (such as those in the work by Keriven (2022), which is limited to linear GNNs [2]) and simple illustrative scenarios. We believe that our empirical approach offers valuable insights that are essential for advancing practical applications in the field.
> > >
> > > **Relevance to the Broader Community**: Our findings offer a different perspective that is vital for researchers and practitioners who rely on these methods for their work. By highlighting the limitations of common practices and challenging prevailing assumptions, our paper contributes to a more accurate and nuanced understanding of oversmoothing in GNNs. This, in turn, can influence future research and the development of more effective methods in graph machine learning. Note that there is a growing body of work investigating oversmoothing, but works typically focus on trying to minimize oversmoothing and avoid questioning if this is actually relevant or useful to improving model performance.
> > >
> > > In summary, we believe that our paper, with its current focus and contributions, is not only relevant but also necessary for the broader graph machine learning community. It addresses key misconceptions and provides empirical evidence that has the potential to improve the current understanding and practices in the field.
> > > We hope our response clarifies the significance of our work, and we remain open to further feedback and discussion.
> > > Thank you for your consideration.
> > > Sincerely,
> > > The Authors
> > >
> > > [1] https://aaai.org/about-aaai/aaai-awards/aaai-conference-paper-awards-and-recognition/
> > >
> > > [2] Keriven, Nicolas. "Not too little, not too much: a theoretical analysis of graph (over) smoothing." Advances in Neural Information Processing Systems 35 (2022): 2268-2281.

---

> ### Comment · Reviewer_yTML · 2023-11-22
>
> I've reviewed the responses from both other reviewers and your comments. I recognize the potential in this work, but it's still in the early stages, with the paper's narrative having shifted, raising additional unanswered questions. It's important to note that reducing oversmoothing doesn't always directly lead to performance enhancements, a point known in the community.
>
> To strengthen the paper, consider these suggestions:
>
> 1. Identify instances where simple methods like dropedge contribute to performance improvement, regardless of oversmoothing considerations.
> 2. Examine cases where such simple methods prove ineffective, outlining reasons for their failure or proposing alternative approaches.
> 3. Conclude the paper by exploring when reducing oversmoothing correlates with performance enhancements and when it
>  does not.
>
> Focusing on these aspects could enhance the paper's impact. I would encourage the authors to pursue some of these directions and submit to the next conference, considering the fact that the rebuttal process is coming to a close, soon.

---

### Official Review · Reviewer_K9Rv · 2023-10-27

**Soundness:** 2 fair
**Presentation:** 3 good
**Contribution:** 2 fair
**Rating:** 3
**Confidence:** 4

**Summary:**

The paper investigates the effects of random dropping approaches for addressing the problem of oversmoothing in GNNs. The paper empirically found that methods like DropEdge and DropMessage which applied randomly and exclusively during training, has limited effect in reducing oversmoothing at test time. Then the paper proposes learning to drop (Learn2Drop) which learns which elements to drop and shows that Learn2Drop is able to successfully mitigate oversmoothing at test time and has better performance than DropEdge and DropMessage.

**Strengths:**

1.	Overall the paper is clearly written and easy to follow.
2.	Investigating the effects of current methods on oversmoothing under both metrics for oversmoothing and model performance is important in order to understand them better and develop more powerful GNNs.
3. Dropping information in an informed way is well-motivated.
4.	If we take NDE as a valid metric to interpret oversmoothing, it is an interesting observation that random dropout methods such as DropEdge and DropMessage applied exclusively during training has little effect on oversmoothing at test time. However, see weakness below why the NDE metric can be not well-defined and not interpretable.

**Weaknesses:**

1. Wrong reference. To the best of my knowledge, Rusch et. al (2022) is not the original paper which first proposes the use of Dirichlet energy to analyze oversmoothing in GNNs. The first paper is Cai and Wang (2020). Please correct the citations in Section 3.1.

2. Missing state-of-the-art literature on oversmoothing. Recently there has been substantial progress made in terms of theoretical understanding of oversmoothing in GNNs such as Keriven (2022) and Wu et al. (2023), where these works rigorously show that there is a “sweet spot” between smoothing and oversmoothing in order for GNNs to perform well for node classification tasks. It is thus not appropriate to claim that "an important takeaway from our work is that the true causes and consequences of oversmoothing are vaguely understood, especially its relationship with overfitting and generalization." Please improve literature review.

3.  No explicit mathematical form for the normalized Dirichlet energy (NDE) and NDE is not a good metric to measure oversmoothing due to at least the three following issues:

- It does not satisfy the criterion proposed for node similarity measure in Rusch et al. (2023), which the authors has relied on as a gold standard for measuring oversmoothing. NDE does not satisfy the criterion proposed in Rusch et al. (2023) because NDE = 0 does not imply all nodes having the same representation vector. It is thus invalid to conclude from NDE that "oversmoothing is still occurring according to the formal definition introduced by Rusch et al. (2023)" from NDE.

- It is obscure to interpret. The reasoning in the paragraph above section 3.2 is entirely heuristic.

- It is not well-defined for 1-d features (all 1-d features have NDE = 0) and cannot be grounded in practice, as it disconnects oversmoothing from model performance.  Consider the following example: suppose that one wants to do a classification task for two group of nodes, where we have one-dimensional, linearly separable features: group 1 all have representation -1, group 2 all have representation 1. Now NDE is 0, which indicates "oversmoothing" according to the paper, but classification result is perfect.

Due to the above reasons, the observation and interpretation drawn upon oversmoothing based on NDE in this paper is not valid.

4. Insufficient empirical evaluation.
- DropEdge and DropMessage can also easily be applied during test time. Although the authors argue that random dropping methods could introduce too much noise during test, the paper does not provide empirical evidence. Please add experiments supporting the claim that using them at test time really reduces model performance.
- In Figure 1, it seems a GNN with skip connections are perfect remedies for oversmoothing, but it is not discussed in the paper and not included as a baseline method in Fig 3. and 4. Please add it as a baseline and discuss.

----
References

Rusch et. al (2022). Graph-coupled oscillator networks. ICLR 2022.

Cai and Wang (2020). A note on over-smoothing for graph neural networks. ICML GRL+ workshop.

Keriven (2022). Not too little, not too much: a theoretical analysis of graph (over)smoothing. NeurIPS 2022.

Wu et al. (2023). A non-asymptotic analysis of oversmoothing in graph neural networks. ICLR 2023.

Rusch et. al (2023). A survey on oversmoothing in graph neural networks. Arxiv.

**Questions:**

1. In Figure 1, it seems that skip connections are perfect remedies for oversmoothing. Why not just go with them?

---

> ### Author Response · Authors · 2023-11-17
> **Response to Reviewer K9Rv**
>
> Dear Reviewer K9Rv,
>
> We thank you for your response and important feedback. We are glad you have pointed out the issues and inconsistencies you found. We have made many changes to the paper in response to your feedback.
>
> > Please improve literature review.
>
> We have revised our literature review of oversmoothing. This has been placed in Section 2.2. Thank you for mentioning the paper by Keriven (2022). Indeed there has been recent work that has tried to analyze the relationship between oversmoothing and generalization from a theoretical perspective. Our work is a contribution from an empirical perspective. One of our conclusions is that oversmoothing is not strictly related to model performance in practice, and minimizing oversmoothing by itself can be undesirable. For example, applying random dropping on the message matrix at test time will reduce oversmoothing according to both MAD and Dirichlet energy – however this *does not result in a useful model*. We believe that this challenges the prior ‘consensus’ in the literature that oversmoothing reduction is desirable, giving new insights from an empirical perspective, and also aligns with the theory work by Kerivan (2022) that the truth is simply ‘reducing oversmoothing is always better’.
>
> > NDE is not a good metric to measure oversmoothing due to at least the three following issues:
>
> We have **removed the normalized Dirichlet energy** (NDE) from the paper, and have **redone our experiments using both standard Dirichlet energy (DE) and mean average distance (MAD)**.
> * In Section 3.1 we have added a clearer explanation of the advantages and drawbacks of each metric. The figures in our paper have also been updated to reflect this. The empirical results on oversmoothing analysis are actually very similar between DE, MAD (and NDE), and the conclusions of our paper remain the same.
> * We note that while the issues faced by MAD (and NDE) are problematic for a theoretical analysis, for our experiments they don’t seem to create a noticeable difference in practice. Our main use case of these metrics is to compare the relative amount of oversmoothing between different GNNs, which is possible with MAD as it exhibits layerwise exponential convergence for high dimensional node representations. While DE has the theoretically sound property that complete smoothing is equivalent to 0, in practice this does not make a significant difference to our empirical observations.
>
> > Although the authors argue that random dropping methods could introduce too much noise during test, the paper does not provide empirical evidence
>
> * We have now included them in Table 1 the results for using DropEdge and DropMessage at test time – we apologize for forgetting to include them in the original submission. We have also taken reviewer yTML’s advice to average the inference results over multiple passes to make this baseline stronger.
>
> > seems a GNN with skip connections are perfect remedies for oversmoothing
>
> GNN+skip connection are now in Figures 1, 3 and 4. It certainly does appear like the ‘perfect remedy’ for oversmoothing like you’ve pointed out – according to both Dirichlet energy and MAD in fact. This is also the case with enabling DropMessage at test time. What this suggests is that while certain methods that perform well are correlated with having lower oversmoothing, it is questionable whether we can justify one model as being better than another by solely using the oversmoothing reduction as our deciding factor. In the case of simply applying DropMessage at test time, this reduces oversmoothing while being detrimental to model performance. In the recent survey by Rush et al (2023), we can see similar results, in which methods such as PairNorm are highly effective in limiting oversmoothing but are outperformed by many other methods. Rusch et al interpret this as the result of some models sacrificing expressive power to reduce oversmoothing. However, we propose the new perspective that a model which *improves performance* may not be strictly related to oversmoothing at all – for instance, DropEdge and DropMessage mainly improve performance by providing a data augmentation effect that benefits training.
>
> Finally, the overall objective of our work wasn’t clearly explained in the original submission. Our main contribution is investigating the effect of random dropping approaches on smoothing, rather than trying to propose a superior method of solving oversmoothing. We’ve revised the abstract, the introduction and conclusion to address this.
>
> Thank you for taking the time to provide us with valuable feedback – we hope you can take a look at our revisions and we would be grateful to hear further comments on areas that can be improved.
>
> Thank you,
>
> The authors.

---

> > ### Comment · Reviewer_K9Rv · 2023-11-21
> > **Follow-up with the authors**
> >
> > Dear authors,
> >
> > Thank you for your response. However, the observation that pure reduction of oversmoothing might sacrifice model performance is not unknown to the community. Despite having done a better literature review during the rebuttal, I feel the paper still needs to go through another round of reframing and reviews to make the contribution more significant.
> >
> > Regarding the problem of residual connections, Wu et al. (2023) (citation provided above) actually have a theoretical analysis on the tradeoff between the reduction of oversmoothing and the reduction in optimal performance and insights about why that is the case. The analysis is much more concrete than "the result of some models sacrificing expressive power to reduce oversmoothing." The authors might find that to be of interest.

---

> > > ### Author Response · Authors · 2023-11-21
> > >
> > > Dear Reviewer K9RV,
> > >
> > > Thank you for your response! Yes -- it is true that the idea that reduction in oversmoothing can be harmful is not a new insight. However, in this work we aimed to focus specifically on random dropping methods -- specifically, our *key contribution* is that DropEdge and DropMessage, do not effectively reduce oversmoothing as previously believed. This observation challenges a key assumption in the field, particularly given the prominence and recent recognition of these methods (e.g., DropMessage being awarded as a distinguished paper at AAAI this year [1]). The main goal of our work is to serve to correct a misconception in current literature on regards to the relationship between these methods specifically oversmoothing. The general observation that pure reduction of oversmoothing might sacrifice model performance is more of a remark which we think reinforces our findings and it is not our intention to claim that it is a novel contribution.
> > >
> > > Regarding your suggestion to explore the theoretical analysis by Wu et al. (2023), we find it highly insightful and relevant. We agree that their work provides a more concrete analysis of the trade-offs involved in reducing oversmoothing, and we plan to incorporate this perspective in future iterations of our research to further contextualize our empirical findings.
> > >
> > > We believe that our work contributes a critical empirical perspective to the ongoing discussion in the field, challenging established notions and guiding future research on graph neural networks. We hope that our response clarifies the intent and significance of our paper.
> > >
> > > We hope this clarification underlines the specific significance of our contribution and would be grateful for any further thoughts or reconsideration you may have on our work.
> > >
> > > Thank you once again for your time and thoughtful feedback.
> > >
> > > Sincerely,
> > >
> > > The authors.
> > >
> > > [1] https://aaai.org/about-aaai/aaai-awards/aaai-conference-paper-awards-and-recognition/

---

> ### Author Response · Authors · 2023-11-21
> **Request for Feedback Before Rebuttal Period Closure**
>
> Dear Reviewer K9Rv,
>
> We would like to kindly request your feedback on the revisions made to our paper -- we have thoroughly addressed the concerns raised in your initial review and made significant changes to the paper.
>
> We would greatly appreciate any additional comments or suggestions you may have before the end of the rebuttal period. We understand the time constraints and appreciate your consideration in providing timely feedback.
>
> Thank you for your valuable contribution to improving our work.
>
> Sincerely,
>
> The authors

---

### Official Review · Reviewer_xrXy · 2023-10-29

**Soundness:** 3 good
**Presentation:** 3 good
**Contribution:** 3 good
**Rating:** 6
**Confidence:** 4

**Summary:**

This paper empirically evaluated the influence of DropEdge and DropMessage on the oversmoothing effect of graph neural networks. The findings reveal that random dropping methods are insufficient in mitigating oversmoothing. The authors then propose a non-random dropping approach that learns which element to drop. This method can be used in both training and testing. Empirically, the proposed method alleviates oversoothing and improves performance accuracy.

**Strengths:**

- The oversmoothing experiments in Section 3 are very thorough. I particularly appreciate section 3.2, where the authors investigate the importance of randomness to DropEdge (Figure 2).
- The proposed information bottleneck approach (Section 4.1) seems to be principled and effective.

**Weaknesses:**

The paper relies on a normalized variant of Dirichlet energy to measure oversmoothing. As the authors themselves pointed out in the paper, however, this metric does not correlate with performance accuracy well. On the one hand, I understand that there is no universally agreed metric for evaluating oversmoothing, and I agree that the normalized Dirichlet energy metric used by the authors is a sensible one. On the other hand, I believe that whether a smoothing effect qualifies as an "oversmoothing" depends on whether the node features give rise to bad final performance (in an extreme case, if constant node features yield the best results, then it is debatable about whether these constant node features are "oversmoothed" or "appropriately smoothed.") Hence, to enhance our understanding of how dropping interacts with oversmoothing effects, I recommend the authors to either:

(i) Plot the normalized Dirichlet energy (x-axis) vs. accuracy (y-axis) frontier for models. This could be informative as the normalized Dirichlet energy alone may not fully reveal oversmoothing in node features.

(ii) Use perhaps more than one metric to evaluate oversmoothing. The Dirichlet energy is one such metric, but other metrics exist. For example, another way of evaluating oversmoothing is to use the influence scores of nodes (Xu et al. 2018), among many other ways.

[1] Representation Learning on Graphs with Jumping Knowledge Networks. Keyulu Xu et al. 2018. ICML.

**Questions:**

Does it make sense to compare test-time DropMessage (where we average different outcomes to reduce variance) with Learn2Drop? As the authors pointed out, while DropMessage can stabilize the Dirichlet energy, applying it at test time can introduce high variance in prediction. However, the variance can be reduced with multiple forward passes, each with a different realization of DropMessage. This will introduce some compute overhead, but seems to be a sensible baseline for Learn2Drop to compare.

---

> ### Author Response · Authors · 2023-11-17
> **Response to Reviewer xrXy**
>
> Dear Reviewer xrXy,
>
> Thank you for taking the time to provide us with feedback and suggestions. We agree with your comments  – oversmoothing is a challenging phenomenon to evaluate. Its relationship with model performance is not as straightforward as one might think – as suggested by our findings. There are many instances where doing simple yet unhelpful things – such as dropping messages at inference time – will apparently solve oversmoothing but fail to improve actual performance. There are also other cases such as with DropEdge, where it may be easy to believe that performance improvement comes from oversmoothing reduction, but in reality may come from elsewhere.
>
> > Use perhaps more than one metric to evaluate oversmoothing.
>
> We have taken your suggestion of using two different metrics of evaluating oversmoothing – we removed the use of normalized Dirichlet energy due to many concerns pointed out by the other reviewers, and now all our experiments are conducted using both the standard Dirichlet energy, as well as MAD. While Dirichlet energy has some good theoretical properties, MAD is also commonly used in the literature and less sensitive to arbitrary scaling of the embeddings. Both metrics lead to similar conclusions – in particular, the analysis of the layer-wise exponential smoothing in section 5.1 is almost identical between Dirichlet energy and MAD. The similarity in the conclusions drawn from both metrics reinforces the soundness of our approach.
>
> > However, the variance can be reduced with multiple forward passes, each with a different realization of DropMessage.
>
> This is a useful suggestion and would serve as a much better baseline than simply applying random dropping at test time. We have modified the table of accuracies in the results section so that the results for DropEdge and DropMessage at test time are done by averaging the model prediction over 10 inference passes. We do witness a small reduction in variance, but the overall results are still poor. Effectively we are averaging over more inference passes and the mean test accuracies are still low. We believe that this provides a more robust comparison and strengthens the validity of our findings.
>
> We highly appreciate your insightful feedback and suggestions – we will be grateful for any further comments.
>
> Thank you,
>
> The authors.

---

> ### Author Response · Authors · 2023-11-21
>
> Dear Reviewer xrXy,
>
> We would like to kindly request your feedback on the revisions made to our paper -- we have thoroughly addressed the concerns raised in your initial review and made significant changes to the paper.
>
> We would greatly appreciate any additional comments or suggestions you may have before the end of the rebuttal period. We understand the time constraints and appreciate your consideration in providing timely feedback.
>
> Thank you for your valuable contribution to improving our work.
>
> Sincerely,
>
> The authors

---

### Official Review · Reviewer_nv43 · 2023-10-31

**Soundness:** 3 good
**Presentation:** 2 fair
**Contribution:** 3 good
**Rating:** 3
**Confidence:** 3

**Summary:**

In this paper, the authors empirically show that existing DropEdge and DropMessage operations have limitations, and propose Learn2Drop to mitigate the oversmoothing issue. Specifically, they propose to optimize a mutual information objective using the information bottleneck principle, and conduct experiments on several datasets to evaluate the effectiveness of Learn2Drop.

**Strengths:**

- The paper studies the oversmoothing issue in GNNs, which is a key issue when applying GNNs.
- Both analysis and emprical results are provided to show the limitations of two existing works.
- Open-sourced code helps improve the reproducibility.
- Empirical experiments are conducted to show the effectiveness of the proposed method.

**Weaknesses:**

- One major concern is from the evaluation part. First, in addition to DropEdge and DropMessage, there are other methods that aim to address the oversmoothing issue. The authors may consider to compare with them. Although the authors mention that the aim of this paper is to isolate and understand the specific impacts of different techniques, it is still encouraged to show to what extent can the proposed method solve the oversmoothing issue compared with the recent, more complex methods. Second, more interpretations related to Table 1 can be provided.

- The authors claim that DropMessage at testing time may introduce a high amount of variance in the model predictions. They may provide more evidence (e.g., some experimental results) on this.

- The writings of the paper can also be improved.

**Questions:**

- Why use Dirichlet energy instead of mean average distance?
- What does DO mean in Table 1?

---

> ### Author Response · Authors · 2023-11-17
> **Response to Reviewer nv43**
>
> Dear Reviewer nv43,
>
> We thank you for your response and important feedback! You highlighted an important point - on the comparison of our Learn2Drop with the latest existing approaches for addressing oversmoothing. We would first like to mention that in the original paper, the main motivation of our work wasn’t clearly addressed and it was misleading. We admittedly overemphasized the significance of ‘reducing oversmoothing’ and would like to highlight that the main novelty of our work is an empirical study and critical look of whether oversmoothing is actually relevant in practice for dropping methods, rather than finding the best way to minimize it.
>
> > The writings of the paper can also be improved.
>
> We have made **significant changes to many sections of the paper** to make its meaning clearer. In particular, the introduction and conclusion sections have been revised to clarify our contributions**. In summary,
>
> * Our primary contribution is analyzing whether random dropping approaches have an effect on oversmoothing, which we show to be not necessarily true since the methods are applied only during training. This suggests that DropEdge for instance, is fundamentally more of a data augmentation tool.
> * We observe that simply dropping random message elements at test time will result in significant oversmoothing reduction, but translates to poor test accuracy. This suggests that minimizing oversmoothing is not necessarily useful.
> * Learn2Drop was motivated by the question of whether we can perform this random dropping at test time in a more informed manner. We don’t aim to beat existing oversmoothing reduction approaches, but rather focus on bringing the potential benefits of existing random dropping methods to test time. Although we observe that Learn2Drop achieves oversmoothing reduction, **this doesn’t prove that reducing oversmoothing is strictly the reason why it performs better**. It is possible that Learn2Drop’s oversmoothing reduction is just a side effect and not the main reason it performs better. This is what we believe is happening for DropEdge. Learn2Drop serves more as a way to give us insight into the relationship between oversmoothing and model performance, as opposed to solely focusing on reducing oversmoothing.
>
> > First, in addition to DropEdge and DropMessage, there are other methods that aim to address the oversmoothing issue. The authors may consider to compare with them.
>
> Following your advice, to make our results more informed, we have added the recent method GraphCon proposed by Rusch (2022) which we treat as a ‘complex’ method of addressing oversmoothing. This is added into Table 1. Although both our approach and GraphCon can effectively prevent the layer-wise exponential behavior of smoothing, there is a still a performance gap between ‘optimally dropping at test time using Learn2Drop’ and GraphCon, which further suggests that it is important to see that oversmoothing is perhaps not the only factor when it comes to making a GNN more accurate. However, an important point is that creating a method that outperforms state of the art methods for reducing oversmoothing is not the purpose / goal of this work, and we don’t expect our approach to result in better accuracy than those methods.
>
> > The authors claim that DropMessage at testing time may introduce a high amount of variance in the model predictions.
>
> We apologize for not having included these results in the original paper – we have also added results into Table 1 showing the performance of models when applying DropEdge and DropMessage at test time. These results are obtained by averaging multiple inference passes to make it a stronger baseline, as suggested by reviewer yTML.. The results are poor and inconsistent, which is likely why the methods were originally proposed to only be used during training.
>
> > Why use Dirichlet energy instead of mean average distance?
>
> The main reason for using Dirichlet energy is that it holds the convenient theoretical property that 0 energy is equivalent to complete oversmoothing (all node representations being identical). However, for our empirical comparisons this property does not make a noticeable difference compared to just using MAD. As other reviewers had concerns about the use of normalized Dirichlet energy, we have removed its use from the paper. Our results were rerun while using standard Dirichlet energy and MAD.
>
> >What does DO mean in Table 1?
>
> That refers to the Dropout method. We have edited that column of the table to make the names clearer.
>
> We hope our revisions to the paper have addressed your concerns and look forward to hearing your thoughts on them.
>
> Thank you,
>
> The authors.

---

> ### Author Response · Authors · 2023-11-21
> **Request for Feedback Before Rebuttal Period Closure**
>
> Dear Reviewer nv43,
>
> We would like to kindly request your feedback on the revisions made to our paper -- we have thoroughly addressed the concerns raised in your initial review and made significant changes to the paper.
>
> We would greatly appreciate any additional comments or suggestions you may have before the end of the rebuttal period. We understand the time constraints and appreciate your consideration in providing timely feedback.
>
> Thank you for your valuable contribution to improving our work.
>
> Sincerely,
>
> The authors

---

> > ### Comment · Reviewer_nv43 · 2023-11-22
> > **Feedback to the authors**
> >
> > Dear authors,
> >
> > Thanks for your response. My questions are partially addressed. Now the paper has been significantly shifted to an empirical study related to the over-smoothing problem. In this sense, the overall contribution is relatively weak and more theoretical analysis is appreciated (especially considering that there are already related analysis out there).

---

> > > ### Author Response · Authors · 2023-11-23
> > >
> > > Dear Reviewer nv43,
> > >
> > > Thank you for your response and for acknowledging the changes we've made to our paper -- we appreciate your perspective on the current focus of our study and understand your viewpoint regarding the need for more theoretical analysis.
> > >
> > > We agree that theoretical analysis is a vital component of research in this field. However, we would like to emphasize that the primary intention of our paper is to address a gap we identified in empirical research related to oversmoothing in GNNs. Our work provides critical insights by empirically evaluating widely-used methods like DropEdge and DropMessage, challenging prevalent assumptions about their effectiveness in reducing oversmoothing. This empirical approach is particularly important given the usage rate of these methods in practical applications.
> > >
> > > We believe our empirical findings contribute significantly to the existing body of knowledge by offering a nuanced perspective that is grounded in practical evaluation. This perspective is crucial for researchers and practitioners in the field who rely on these methods for real-world applications.
> > >
> > > That said, we acknowledge the importance of theoretical analysis and are committed to integrating more theoretical perspectives into our work. To this end, we are exploring ways to augment our empirical findings with theoretical insights, possibly drawing from existing analyses while providing a unique angle based on our empirical results.
> > >
> > > We hope this response clarifies the rationale behind our empirical focus and our commitment to enhancing the paper with theoretical insights.
> > >
> > > Thank you once again for your feedback.
> > >
> > > Sincerely,
> > >
> > > The authors.

---

### Author Response · Authors · 2023-11-17
**Thank you for your reviews**

Dear reviewers,

We would like to thank you for your time and effort in reading our paper and providing us with your reviews. Overall we have carefully considered all of your advice and made significant changes to the paper (the edits / revised sections have been highlighted in blue). Here is a summary of everything we have changed.
1. The normalized Dirichlet energy, as pointed out by reviewers K9Rv and yTML, has unaddressed flaws. We have decided to remove it from the paper. Instead, we redid our experiments using both standard Dirichlet energy and MAD and updated all figures. In Section 3.1 we now have a discussion on the benefits and limitations of each metric. The overall findings and conclusions are actually very similar between all three metrics, suggesting that their theoretical differences may not matter for the empirical study we have conducted.
2. We have revised and clarified the main objectives, contributions and takeaways of our work. We believe that in the original submission, the main motivation of our work wasn’t clearly addressed and was slightly misleading. Our contribution is an empirical analysis of the effect of random dropping on oversmoothing – we examine in detail whether random dropping actually contributes to reducing oversmoothing, and how this relates to the GNN’s performance. We propose Learn2Drop in response to the ineffectiveness of random dropping at test time. Its main motive is to help us better understand the relationship between oversmoothing and performance in the context of dropping approaches, rather than to compete with more recently proposed methods that strictly target oversmoothing.
3. As suggested by reviewer K9Rv, we have revised our literature review of oversmoothing in Section 2.2. We have mentioned and compared our work to Keriven (2022)’s theoretical analysis of oversmoothing and model performance. Our overall findings are an empirical contribution, suggesting that in practice, reducing oversmoothing is not strictly beneficial to performance. This aligns with Keriven (2022)’s theoretical conclusion that seeking to minimize oversmoothing is not necessarily the best choice. We believe that our empirical contributions are meaningful, as in many works, the idea that minimizing oversmoothing is a desirable goal is often assumed without justification.
4. We have now included in Table 1 the results for using DropEdge and DropMessage at test time, which we claimed to be poor but forgot to include in the original submission as mentioned by reviewer nv43. Following xrXy’s suggestion, this is done by averaging multiple forward passes rather than taking a single forward pass. We have also included GraphCon by Rusch (2022) to add context to the performance of L2D.
5. We have conducted some extra experiments. Following reviewer yTML’s comment that oversmoothing within the same class may not be relevant, we now additionally analyze the relationship between model performance and oversmoothing across *different* classes in Section 3.3 by measuring the Dirichlet energy and MAD only on pairs of nodes with different labels.
6. We have included an ‘experimental setup’ appendix in response to reviewer yTML’s concern on the lack of reproducibility.
7. Sections 4.1 and 4.2 have been largely rewritten as they were previously quite hard to follow.

We hope that the revisions to the paper have addressed the concerns you have mentioned. We very much look forward to any further feedback you may have.

Thank you,

The authors.

---

### Meta-Review · Area_Chair_BFHS · 2023-12-06

**Metareview:**

The paper investigates the issue of over-smoothing in GNNs and evaluates the limitations of existing methods, DropEdge and DropMessage, which randomly drop information during training. The authors propose a new method called Learn2Drop, which learns which elements to drop non-randomly. They use the normalized Dirichlet energy metric to measure over-smoothing and conduct experiments to show that Learn2Drop effectively mitigates over-smoothing and improves performance compared to DropEdge and DropMessage.

Strengths of the paper are the following:
* The paper conducts a comprehensive evaluation of over-smoothing using various metrics and experiments, providing a detailed analysis of the limitations of existing methods.
* The paper addresses an important problem in GNNs and proposes a novel solution, Learn2Drop, which shows promising results in mitigating over-smoothing.

Weaknesses of the paper:
* The paper primarily compares Learn2Drop with DropEdge and DropMessage but does not compare it with other recent, more complex methods that address over-smoothing. This comparison could provide a more comprehensive assessment of Learn2Drop's effectiveness.
* The use of NDE as a metric to measure over-smoothing is questioned by one reviewer due to its limitations and lack of correlation with performance accuracy. The paper may benefit from considering alternative metrics or providing additional evidence supporting the choice of NDE.
* The paper lacks references to recent works on over-smoothing in GNNs, which could provide valuable context and insights.

The paper makes a valuable contribution to the understanding of oversmoothing in GNNs and proposes a promising solution in the form of Learn2Drop. However, it could benefit from a more extensive evaluation, considering alternative metrics, including recent literature.

**Justification For Why Not Higher Score:**

Considering these concerns and weaknesses highlighted by the reviewers, the current recommendation is "reject" because the paper needs substantial revisions and improvements in methodology, evaluation metrics, literature review, and experimental design to address the reviewers' concerns.

**Justification For Why Not Lower Score:**

N/A.

---

### Decision · Program_Chairs · 2024-01-16

Reject